# QuickMerge++: Token Merging with Autoregressive Prior

**Dong Liu** [1]   **Yanxuan Yu** [2]

## Abstract

As generative models scale to larger inputs across language, vision, and video domains, the cost of token-level computation has become a key bottleneck. While prior work suggests that only a subset of tokens significantly influence downstream predictions, most token selection methods are static, modality-specific, or incompatible with autoregressive generation. In this paper, we propose QuickMerge, a lightweight token merging framework designed for efficient next-token prediction.

QuickMerge dynamically selects a reduced number of tokens based on attention norm magnitude, guided by an entropy-based budget estimator. To preserve autoregressive compatibility, we introduce a lightweight transformer prior trained over the merged token sequence. By combining semantic salience estimation, flexible token budgets, and AR alignment, QuickMerge enables accurate generation with fewer tokens.

We evaluate QuickMerge across multi-modality domains, demonstrating consistent improvements in compute-accuracy tradeoffs. Specifically, QuickMerge reduces token counts sustantially while matching as well as exceeding the performance of learned tokenizers and fixed-patch baselines. Our code implementation can be found at https://github.com/NoakLiu/QuickMerge.

## 1. Introduction

Large-scale generative models have achieved remarkable success across language, vision, and multimodal domains. However, their inference and training cost grows linearly with the number of input tokens, creating a fundamental bottleneck when processing long-context sequences or high-resolution inputs. This challenge is particularly acute in autoregressive (AR) generation, where every input token participates in recurrent attention and prediction.

Recent efforts have shown that not all tokens contribute equally to model performance (Ryoo et al., 2021; Kim et al., 2024). A small subset of salient tokens typically dominates the output prediction, suggesting the possibility of selective computation. Yet, most existing token pruning or merging strategies rely on static heuristics, pre-defined token layouts, or non-autoregressive assumptions. These approaches either degrade generation quality or fail to integrate with decoder-only AR models.

To address this, we propose **QuickMerge++**, a lightweight and autoregressive-compatible framework for inference-time token reduction. Our method introduces three key innovations: (1) an entropy-aware mechanism to estimate local input complexity and determine dynamic token budgets; (2) a saliency-guided token merging strategy based on mass-weighted averaging of semantically redundant tokens; and (3) a compact autoregressive prior trained over the merged token sequences to ensure compatibility with downstream generation.

QuickMerge++ is modality-agnostic and plug-and-play: it operates on frozen encoder outputs and applies to text, image, and video inputs alike. Unlike fixed-length quantization or manual patching, it enables adaptive compression conditioned on semantic density. Empirical results across multiple benchmarks demonstrate that QuickMerge++ achieves up to 3× token reduction with minimal or no drop in generation quality.

Our contributions are summarized as follows:

- We identify the mismatch between token-level redundancy and autoregressive decoding, motivating a saliency-aware merging strategy.

- We propose QuickMerge++, a general-purpose framework that combines entropy-based budgeting, norm-weighted merging, and AR-compatible modeling.

- We validate our approach across modalities, showing consistent improvements in efficiency–accuracy tradeoffs on long-context generation tasks.

---

[1]Yale University [2]Columbia University. Correspondence to: Dong Liu <dong.liu.dl2367@yale.edu>.

*Proceedings of the 42nd International Conference on Machine Learning*, Vancouver, Canada. PMLR 267, 2025. Copyright 2025 by the author(s).

# 2. Related Work

## 2.1. Tokenization for Vision, Language, and Video

Tokenization forms the foundation for modern generative models across modalities. In vision, ViT (Dosovitskiy et al., 2020) and VideoMAE (Tong et al., 2022) adopt patch-based tokenization, splitting inputs into fixed-size grids. While simple and effective, these approaches produce a rigid number of tokens regardless of content complexity.

Quantization-based methods like VQ-VAE (Van Den Oord et al., 2017) and its extensions introduce discrete latent representations by learning a codebook. However, they suffer from fixed vocabulary size, mode collapse, and are difficult to align with autoregressive generation.

In language modeling, token-free methods such as ByT5 (Xue et al., 2022) and Charformer (Tay et al., 2021) eliminate subword tokenization altogether, directly modeling raw byte or character sequences. Besides, there are also contemporary toolbox such as LLMEasyQuant (Liu & Yu, 2025b) and TensorRT (Davoodi et al., 2019) for easy-to-use quantization on languange models. While conceptually elegant, these approaches often require deeper models or extensive training to match subword-level performance.

## 2.2. Dynamic and Learned Token Selection

Dynamic tokenization aims to adaptively select salient inputs based on task or context. TokenLearner (Ryoo et al., 2021) introduces learned soft-attention pooling to produce a compact set of tokens for vision tasks. Despite its flexibility, it lacks autoregressive alignment and enforces a fixed output size.

LARP (Wang et al., 2024) addresses autoregressive generation by introducing a learned prior over discrete tokens for video. It aligns tokenization with AR objectives but is specific to video and assumes an encoder-decoder structure.

Recent work has explored dynamic and saliency-based token compression strategies. Nawrot et al. (Nawrot et al., 2023) propose adaptive token compression in ViTs based on local feature importance. Thiombiano et al. (Thiombiano et al., 2025) present entropy-aware routing for efficient generative modeling. Compared to these, QuickMerge++ introduces a modular, entropy-guided strategy with explicit autoregressive compatibility.

QuickMerge draws inspiration from these efforts but differs in three key ways: it supports variable token budgets, operates at inference time with no retraining, and integrates seamlessly with any backbone.

## 2.3. Efficient Learning and System Optimization

In parallel with tokenization and compression research, recent advances in efficient learning systems highlight orthogonal strategies for accelerating inference and improving scalability.

HADES (Yang et al., 2024) introduces hardware-accelerated speculative decoding tailored for large language models, enabling low-latency generation via speculative sampling and early validation. FastCache (Liu et al., 2025a) accelerates Diffusion Transformers by replacing costly quadratic cache lookups with a learnable linear surrogate, delivering up significant end-to-end speed-ups. System-level studies such as (Jin & Yang, 2025; Ji & Luo, 2025) explore elastic scaling and self-healing inference pipelines in cloud-based environments, crucial for production-scale deployment of autoregressive models.

From the learning algorithm side, MT2ST (Liu & Yu, 2024) and model fusion frameworks (Liu et al., 2025b) aim to unify task-agnostic and task-specific capabilities under minimal retraining. These directions are complementary to QuickMerge, which focuses on inference-time adaptation without altering model weights.

Federated and collaborative learning efforts such as AppFL (Li et al., 2024) and privacy-preserving cloud systems (Luo & Ji, 2025) push the boundary of distributed and privacy-aware inference. Their compression-aware infrastructure can benefit from adaptive token reduction modules like QuickMerge to reduce bandwidth and latency.

More broadly, recent works on data augmentation (Yang et al., 2025; Liu & Jiang, 2024), model compression (Liu, 2024), and retrieval acceleration (Liu et al., 2024; Liu & Yu, 2025a) all signal the growing importance of plug-and-play modules that integrate with large models without retraining.

QuickMerge situates itself within this efficient learning paradigm by offering a unified, lightweight, and modality-agnostic token reducer that complements both architectural and system-level optimizations.

## 2.4. Token Merging and Compression

Recent studies have explored reducing token count via pruning and merging. DynamicViT (Rao et al., 2021) prunes low-importance tokens progressively throughout layers. TokenFusion (Kim et al., 2024) merges nearby visual tokens with similar content to accelerate ViT models.

However, these methods typically require training-time modifications or degrade performance in generation settings. In contrast, QuickMerge provides a plug-and-play module that performs token merging based on entropy and norm-based scores, and maintains compatibility with autoregressive decoding through a lightweight transformer prior.

Our method can be interpreted as a synthesis of token compression, semantic selection, and autoregressive alignment, suitable for text, image, and video modalities under a unified framework.

## 3. Methodology

We propose **QuickMerge++**, a modality-agnostic token compression framework that accelerates generative modeling by reducing sequence length while preserving autoregressive compatibility. QuickMerge++ consists of four main stages: (1) multi-scale entropy-aware saliency estimation, (2) differentiable token merging with structural weighting, (3) bidirectional autoregressive alignment, and (4) compression-aware fidelity control.

### 3.1. Problem Setup

Let $X \in \mathbb{R}^{B \times N \times D}$ denote token embeddings from a frozen encoder (e.g., ViT, BERT, VideoMAE). Our goal is to construct a compressed sequence $\tilde{X} \in \mathbb{R}^{B \times K \times D}$ with $K \ll N$, suitable for downstream left-to-right decoding by an autoregressive model $f_\theta$. The compression function $g$ must satisfy:

$$\tilde{X} = g(X), \quad \text{such that } K \leq K_{\max}, \quad f_\theta(\tilde{X}_{\leq t}) \approx \tilde{X}_{t+1}$$

### 3.2. Stage 1: Multi-Scale Entropy-Aware Saliency

For each token $x_i$, we estimate its contextual importance using attention entropy across $L$ transformer layers. Let $A^{(l)} \in \mathbb{R}^{N \times N}$ denote the attention matrix at layer $l$:

$$A^{(l)} = \text{softmax}\left(\frac{X^{(l)}(X^{(l)})^\top}{\sqrt{D}}\right) \quad (1)$$

$$H_i^{(l)} = -\sum_{j=1}^{N} A_{ij}^{(l)} \log A_{ij}^{(l)} \quad (2)$$

The average normalized entropy across layers defines the saliency score:

$$s_i = \frac{1}{L} \sum_{l=1}^{L} \text{Normalize}(H_i^{(l)})$$

This entropy-based signal favors sharp attention tokens with low uncertainty, identifying key semantic contributors.

### 3.3. Stage 2: Differentiable Token Merging

We perform soft selection of salient tokens via Gumbel-softmax:

$$\pi_i = \frac{\exp((s_i + g_i)/\tau)}{\sum_j \exp((s_j + g_j)/\tau)}, \quad g_i \sim \text{Gumbel}(0, 1) \quad (3)$$

$$M_i \sim \text{GumbelSoftmax}(\pi_i) \quad (4)$$

The token mass used for merging is defined as:

$$\tilde{m}_i = M_i \cdot s_i + (1 - M_i) \cdot \epsilon, \quad \epsilon \ll 1$$

Tokens are grouped into clusters $\mathcal{G}_k$ (e.g., using KNN or agglomerative cosine clustering). Each merged token is a saliency-weighted average:

$$\tilde{x}_k = \sum_{j \in \mathcal{G}_k} \frac{\tilde{m}_j x_j}{\sum_{j' \in \mathcal{G}_k} \tilde{m}_{j'}}$$

### 3.4. Stage 3: Bidirectional AR Prior Alignment

After token compression, we obtain a sequence of merged tokens $\tilde{X} = [\tilde{x}_1, \tilde{x}_2, \ldots, \tilde{x}_K]$, where $K \ll N$. To ensure that this compressed representation can be used for autoregressive generation, we introduce a bidirectional AR training objective.

Let $f_\rightarrow$ denote the forward autoregressive decoder and $f_\leftarrow$ the backward decoder. At training time, we jointly train both directions to predict the next (or previous) token in the compressed sequence, thus preserving the internal temporal consistency of the original input.

**Forward prediction.** At each position $t \in \{1, 2, \ldots, K-1\}$, the forward decoder predicts the next token embedding $\tilde{x}_{t+1}$ based on the prefix context:

$$\mathcal{L}_{\text{forward}} = \sum_{t=1}^{K-1} \|f_\rightarrow(\tilde{x}_1, \tilde{x}_2, \ldots, \tilde{x}_t) - \tilde{x}_{t+1}\|^2$$

**Backward prediction.** Similarly, the backward decoder predicts the previous token based on the suffix:

$$\mathcal{L}_{\text{backward}} = \sum_{t=2}^{K} \|f_\leftarrow(\tilde{x}_K, \tilde{x}_{K-1}, \ldots, \tilde{x}_t) - \tilde{x}_{t-1}\|^2$$

**Combined objective.** The final autoregressive training loss combines both directions:

$$\mathcal{L}_{\text{AR}} = \mathcal{L}_{\text{forward}} + \mathcal{L}_{\text{backward}}$$

This bidirectional loss encourages the compressed token sequence to retain enough structural information to allow fluent left-to-right generation, while also preserving temporal coherence in reverse (e.g., useful for sequence-level tasks or reversed decoding). Notably, only the forward AR decoder $f_\rightarrow$ is used during inference.

### 3.5. Stage 4: Compression-Aware Fidelity Constraint

To quantify compression impact, we define the cumulative retained norm:

$$\gamma = \frac{\sum_{i \in \text{TopK}(\|x_i\|)} \|x_i\|}{\sum_{i=1}^{N} \|x_i\|}$$

Assuming norm correlates with informativeness, the following fidelity bound holds:

$$\|X - \tilde{X}_{\text{pad}}\|_F^2 \leq (1 - \gamma)^2 \cdot \|X\|_F^2$$

where $\tilde{X}_{\text{pad}}$ pads $\tilde{X}$ back to $N$ tokens. This provides an upper bound on representation error.

### 3.6. Inference Pipeline

---
**Algorithm 1** QuickMerge++ Inference Pipeline
---
**Require:** Token embeddings $X \in \mathbb{R}^{N \times D}$, AR model $f_\theta$, temperature $\tau$, max token count $K_{\max}$
1: **for** layer $l = 1$ to $L$ **do**
2: $\quad A^{(l)} \leftarrow \text{softmax}(X^{(l)}(X^{(l)})^\top / \sqrt{D})$
3: $\quad H_i^{(l)} \leftarrow - \sum_j A_{ij}^{(l)} \log A_{ij}^{(l)}$
4: **end for**
5: $s_i \leftarrow \frac{1}{L} \sum_l \text{Normalize}(H_i^{(l)})$
6: $g_i \sim \text{Gumbel}(0, 1), \quad \pi_i \leftarrow \text{softmax}((s_i + g_i)/\tau)$
7: $M_i \leftarrow \text{GumbelSoftmaxSample}(\pi_i)$
8: $\tilde{m}_i \leftarrow M_i \cdot s_i + (1 - M_i) \cdot \epsilon$
9: $\{\mathcal{G}_1, \ldots, \mathcal{G}_K\} \leftarrow \text{Cluster}(X, \tilde{m}, K = K_{\max})$
10: **for** each cluster $\mathcal{G}_k$ **do**
11: $\quad \tilde{x}_k \leftarrow \sum_{j \in \mathcal{G}_k} \frac{\tilde{m}_j x_j}{\sum_{j' \in \mathcal{G}_k} \tilde{m}_{j'}}$
12: **end for**
13: $\tilde{X} \leftarrow \{\tilde{x}_1, \ldots, \tilde{x}_K\}$
14: **for** $t = 1$ **to** $K-1$ **do**
15: $\quad \hat{x}_{t+1} \leftarrow f_\theta(\tilde{x}_1, \ldots, \tilde{x}_t)$
16: **end for**
17: **return** Compressed AR-compatible sequence $\tilde{X}$
---

### Discussion

QuickMerge++ differs from recent dynamic compression methods (Nawrot et al., 2023; Thiombiano et al., 2025) by maintaining full compatibility with left-to-right decoding. Our entropy-guided saliency combines local and global cues, and the fidelity-bound analysis provides a practical signal for compression budget calibration. The entire pipeline is modular, lightweight, and training-free for the encoder, enabling plug-and-play integration across modalities.

## 4. Experiments

We conduct extensive experiments to evaluate **Quick-Merge++** across *text*, *image*, and *video* modalities. Our experiments are designed to answer the following:

**Q1.** Can QuickMerge++ reduce token count while preserving generation quality?

**Q2.** What is the contribution of each component: entropy-based budgeting, saliency-guided merging, and autoregressive (AR) prior?

**Q3.** How efficient is QuickMerge++ in terms of runtime, memory, and compression trade-offs?

### 4.1. Setup and Metrics

**Datasets.** We evaluate across modalities: WikiText-103 (Merity et al., 2016), ImageNet-1K (Deng et al., 2009), and UCF101 (Soomro et al., 2012), and extend to long-context tasks: BookSum (Kryściński et al., 2021), Ego4D-NLQ (Grauman et al., 2022).

**Metrics.** We use task-specific quality metrics: PPL (text), accuracy (image), FVD (video), ROUGE / mAP (long-context). Efficiency is assessed via compression rate ($K/N$), decoding latency, and KV memory cost.

**Model.** A 6-layer Transformer decoder (hidden size 512) is trained with fixed pretrained encoders (BERT, ViT, Video-MAE). QuickMerge++ dynamically selects tokens with entropy threshold $\alpha = 0.45$ and norm masking. Results are averaged over 3 seeds.

### 4.2. Overall Performance (Q1)

*Table 1.* QuickMerge++ improves performance while compressing input tokens. Values are mean ± std over 3 runs.

| Method | PPL ↓ | Acc ↑ | FVD ↓ | CompRate ↑ |
|---|---|---|---|---|
| Fixed Patches | 21.4 | 76.2 | 108.4 | $1.00\times$ |
| VQ-VAE | 19.8 | 74.9 | 97.2 | $1.31 \times \pm 0.02$ |
| Token-Free | 17.9 | 76.3 | 89.7 | $2.08 \times \pm 0.05$ |
| TokenLearner | 18.6 | 77.0 | 94.1 | $1.83 \times \pm 0.04$ |
| **QuickMerge++** | **17.1** | **78.1** | **85.6** | $\mathbf{2.37 \times \pm 0.06}$ |

QuickMerge++ consistently achieves stronger results with fewer tokens. On average, it yields $2.37\times$ compression with up to 4.3% relative improvement in accuracy or quality, confirming Q1. The token budget adapts to sequence complexity—e.g., longer sequences yield higher savings (Table 3).

### 4.3. Component Analysis (Q2)

We ablate each component in QuickMerge++:

*Table 2.* Component-wise ablation (averaged across all tasks).

| Variant | PPL ↓ | Acc ↑ | FVD ↓ |
|---|---|---|---|
| Full Model | **17.1** | **78.1** | **85.6** |
| – Entropy Budgeting | 18.4 | 76.5 | 91.0 |
| – AR Prior | 17.9 | 76.9 | 88.3 |
| – Norm Masking | 18.7 | 75.2 | 93.5 |

Each module is necessary: removing entropy control

increases overcompression variance; removing the AR prior impairs temporal alignment; removing norm masking merges semantically irrelevant tokens. This supports Q2.

### 4.4. Efficiency and Scaling (Q3)

We benchmark runtime and memory usage in autoregressive generation. Experiments are run on an NVIDIA A100 (batch size 32):

Table 3. Efficiency metrics of QuickMerge++.

| Metric | Baseline | QuickMerge++ | Rel. Change |
|---|---|---|---|
| Latency (ms) | 6.3 | 4.1 | $-34.9\%$ |
| KV Memory (MB) | 1120 | 412 | $-63.2\%$ |
| Tokens (mean) | 128 | 54 | $-57.8\%$ |

QuickMerge++ significantly reduces decoding latency and memory load with only minor preprocessing overhead ($+1.6$ ms). These results confirm Q3.

### 4.5. Extended Benchmarks

**Long-Context.** On BookSum (text summarization) and Ego4D-NLQ (video QA), QuickMerge++ reduces token count by $2.4$–$2.7\times$ while improving ROUGE-L and mAP:

Table 4. QuickMerge++ results on long-context benchmarks. It improves quality metrics while significantly reducing token count.

| Task | Metric | Baseline | QuickMerge++ | Improve |
|---|---|---|---|---|
| | Quality ($\uparrow$) | | | |
| BookSum | ROUGE-L | 36.2 | **37.1** | $+0.9$ |
| Ego4D-NLQ | mAP@0.5 | 38.6 | **40.3** | $+1.7$ |
| | Token Count ($\downarrow$) | | | |
| BookSum | Tokens | 4096 | **1682 $\pm$ 33** | $-58.9\%$ |
| Ego4D-NLQ | Tokens | 2048 | **821 $\pm$ 25** | $-59.9\%$ |

**Cross-Task Transfer.** We evaluate QuickMerge++ (no retraining) on MSCOCO, Something-Something-v2, TVQA, and QASPER. Compression rates fluctuate based on input entropy:

Table 5. Generalization across unseen tasks.

| Task | Metric $\uparrow$ | QMerge++ / Baseline | Compression |
|---|---|---|---|
| MSCOCO | BLEU-4 | 36.0 / 35.7 | $2.21 \pm 0.07$ |
| SSv2 | Accuracy | 53.8 / 53.1 | $2.06 \pm 0.05$ |
| TVQA | QA Acc | 71.5 / 71.2 | $1.92 \pm 0.06$ |
| QASPER | F1 | 76.4 / 75.9 | $2.35 \pm 0.04$ |

Compression adapts to complexity without hurting performance—validating QuickMerge++ as a plug-in token reducer.

### 4.6. Conclusion

QuickMerge++ provides a general, adaptive, and efficient token merging strategy across domains. By leveraging entropy-guided budgeting, norm-based saliency, and autoregressive priors, it reduces token count by $2.0$–$2.5\times$ while preserving or improving task performance and reducing compute cost. All three research questions (Q1–Q3) are affirmatively answered.

## 5. Conclusion

We present **QuickMerge++**, a lightweight and modality-agnostic framework for token reduction in generative modeling. By integrating entropy-aware token budgeting, saliency-guided merging, and autoregressive prior alignment, Quick-Merge++ provides a principled solution to the growing inefficiency of dense token sequences. Extensive experiments across text, image, and video domains demonstrate that QuickMerge++ achieves significant token compression while maintaining or improving generation quality. Furthermore, it generalizes effectively across tasks and domains without retraining. These results suggest that adaptive token merging—grounded in semantic salience and generative compatibility—can serve as a key building block in the next generation of efficient autoregressive systems.

In future work, we plan to integrate QuickMerge++ with streaming decoders, long-context memory modules, and multi-agent generative systems to further expand its scalability and applicability.

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

# Theoretical Analysis of Entropy-Based Token Merging

## A. Normalized Entropy Drop (NED)

We define the *Normalized Entropy Drop* (NED) as a saliency signal for identifying non-informative tokens:

$$\text{NED}_i = \frac{H_i^{\text{in}} - H_i^{\text{out}}}{H_i^{\text{in}} + \delta}$$

where $H_i^{\text{in}}$ denotes the average incoming attention entropy of token $i$ and $H_i^{\text{out}}$ the outgoing entropy, and $\delta > 0$ is a small constant to ensure numerical stability.

**Interpretation.** Tokens with high incoming entropy but low outgoing entropy tend to absorb dispersed attention but contribute little to other tokens—a signal of redundancy.

**Monotonicity Lemma.** Under a simplified isotropic softmax attention, we show:

$$\text{NED}_i \propto \frac{\text{Var}(A_{\cdot i})}{\mathbb{E}[A_{\cdot i}]} - \text{Var}(A_{i \cdot})$$

This implies that higher NED corresponds to higher incoming attention inconsistency and lower outgoing importance, motivating removal or merging.

## B. Stability Across Layers

We analyze the propagation of NED over attention layers. Let $H_i^{(l)}$ denote the entropy at layer $l$, then:

$$\left| \text{NED}_i^{(l)} - \text{NED}_i^{(l+1)} \right| \leq \mathcal{O}(\|X^{(l)} - X^{(l+1)}\|_F)$$

This suggests that NED is stable under small representation drifts and provides consistent token saliency estimates across layers.

## C. Attention Variance and Merge Risk

Let $x_i$ be a candidate for merging and define $\sigma_i^2 = \text{Var}(A_{\cdot i})$. Then under Gaussian attention models, we derive:

$$\mathbb{P}[x_i \text{ contributes significantly}] \leq \exp\left( -\frac{\text{NED}_i^2}{2\sigma_i^2} \right)$$

which supports a probabilistic guarantee that low-NED tokens are unlikely to be critical for generation.

# Complexity and Error Propagation Bounds

## A. Complexity Analysis

Let the original sequence length be $N$ and merged length be $K$.

**Token Reduction.** If QuickMerge++ merges tokens in $G$ groups with average size $|G| = N/K$, then the resulting self-attention cost reduces from:

$$\mathcal{O}(N^2 D) \quad \rightarrow \quad \mathcal{O}(K^2 D) = \mathcal{O}\left( \left( \frac{N}{|G|} \right)^2 D \right)$$

showing up to $\mathcal{O}(|G|^2)$ speedup.

**Merge Overhead.** Gumbel-softmax and saliency computation scale linearly in $N$, i.e., $\mathcal{O}(ND)$.

## B. Reconstruction Error Upper Bound

Let $X = [x_1, \ldots, x_N] \in \mathbb{R}^{N \times D}$ and merged tokens be $\tilde{X} = [\tilde{x}_1, \ldots, \tilde{x}_K]$. Assume merge groups $\mathcal{G}_i$ with saliency weights $m_j$. Then:

$$\left\| X - \tilde{X}_{\text{pad}} \right\|_F^2 \leq \sum_{i=1}^{K} \sum_{j \in \mathcal{G}_i} m_j \| x_j - \tilde{x}_i \|^2$$

Using triangle inequality and Cauchy-Schwarz:

$$\leq (1 - \gamma)^2 \cdot \|X\|_F^2, \quad \text{where } \gamma = \frac{\sum_{j \in \text{TopK}(\|x_j\|)} \|x_j\|}{\sum_{i=1}^{N} \|x_i\|}$$

**Interpretation.** This implies that the total information loss is bounded by how much salient norm mass is retained.

## C. Worst-Case Prediction Divergence Bound

Let $f$ be an $L$-Lipschitz autoregressive model. Let $z$ and $\tilde{z}$ be the full and merged sequences, then:

$$\mathbb{E}[\|f(\tilde{z}_{\leq t}) - f(z_{\leq t})\|^2] \leq L^2 \cdot \mathbb{E}[\|\tilde{z}_{\leq t} - z_{\leq t}\|^2]$$

implying that downstream prediction error grows at most linearly with token merge distortion.

