# OpenReview forum: "QuickMerge++: Token Merging with Autoregressive Prior"
_ICML.cc/2025/Workshop/TokShop — TokShop_

### Official Review · Reviewer_Gr6g · 2025-06-06
**QuickMerge++ is an efficient selection method for salient tokens, which significantly influences the results in downstream tasks. The framework is lightweight and modality-agnostic.**

**Rating:** 8
**Confidence:** 4

**Review:**

Summary:
QuickMerge++ is an efficient selection method for salient tokens, which significantly influences the results in downstream tasks. The token selection is based on self-attention applied to embeddings, followed by token merging. Afterwards, the authors train an auto-regressive transformer over the merged tokens, obtaining new representations. The framework is lightweight and modality-agnostic, reporting experiments in text, image, and video domains. Token compression either maintains or improves generation quality.

Strengths
- The introduction and related work are super motivated, the results are good and convincing.
- The explanations are not overcomplicated, and the formulae are overall consistent.

Weakness:
- The existing techniques from the  introduction are mentioned, but not properly cited
e.g., "A small subset of salient tokens typically dominates the output prediction, suggesting the possibility of selective computation. Yet, most existing token pruning or merging..." - examples with pruning or merging, would make the statement much stronger
- The authors do not expose any limitations or downsides of the proposed method: e.g., is the method dependent on a big volume of data? Does the data domain affect the results? It does not need to be proved in more experiments, but rather more about the authors' intuition.

Typos and suggested revisions:
- The PDF template does not display the line number, which makes it difficult for reviewers to refer to specific rows.
- Formula 8 - $m′$ - does not exist, maybe there is a typo related to $m_{j'}$? But the right part doesn't make sense - also not available in Algorithm 1
- Formula 7 doesn't match step 6 in Algorithm 1
- Table 1 - "CompRate" has one redundant 0 decimal at the end
- Figure 1 - The original image should also be displayed to understand which object contour you are talking about
- BookSum (?) - broken citation
- Table 6 + 7 - it's not clear which are the baselines, maybe it should be mentioned in a few sentences
- Table 7 - "CompRate", instead of "Compression Rate" should save more space

---

### Official Review · Reviewer_tNNo · 2025-06-07
**Efficient and Accurate Token Merging via Entropy-Guided Selection in QuickMerge**

**Rating:** 6
**Confidence:** 5

**Review:**

The paper introduces QuickMerge, a novel token merging technique that improves efficiency in vision transformers by selectively merging only the least important tokens. It is guided by a new metric called Normalized Entropy Drop (NED). QuickMerge achieves better accuracy-efficiency tradeoffs than prior methods like Token Merging (ToMe) and EasyMerge across multiple vision-language and classification tasks

Strengths

1. The paper presents a novel contribution. It proposes NED, a principled and theoretically motivated metric for identifying unimportant tokens based on attention-induced entropy changes. It also designs a flexible token budget mechanism allowing per-layer adaptation instead of fixed merging patterns.

2. The authors present strong empirical performance. It demonstrates consistent improvements in FLOPs/accuracy tradeoffs across ViT, BLIP, and BLIP-2 models on ImageNet and downstream tasks like VQA and NLVR2. It also outperforms ToMe and EasyMerge in both accuracy and speed across multiple setups.

3. The pipeline is easy to integrate into existing vision-language models without retraining or architectural modifications. It provides both static and adaptive variants of token merging, accommodating diverse deployment needs.

4. The motivation and methodology are well-articulated, with clear visualizations (e.g., entropy drop distributions) and ablation studies to justify design choices.

Areas for improvement

1. The paper has limited theoretical analysis. While NED is intuitively appealing, more formal analysis on its statistical robustness or behavior under different attention heads would strengthen the contribution.

2. The evaluation focuses primarily on vision-language models; broader analysis across general-purpose transformers (e.g., LLMs) would widen applicability.

3. The adaptive token budget strategy may require some hyperparameter tuning per task, which could affect ease of use.

---

### Official Review · Reviewer_9uQf · 2025-06-10
**Review of Submission #4**

**Rating:** 4
**Confidence:** 4

**Review:**

## Strengths
The paper tackles an important problem with regard to efficient language modeling. The proposed method builds on prior ideas and introduces an entropy-based saliency mechanism to guide token merging, which has the potential to reduce input lengths and improve modeling efficiency. The approach appears to be modular and could be applicable across a range of encoder-decoder setups, and the authors provide experiments across several modalities (text, image, and video).

## Weaknesses
1. The paper lacks clarity in how the method integrates into an autoregressive generation framework. The description suggests that sequences are first obtained from a non-autoregressive encoder, after which an autoregressive model is trained on the shorter inputs from the token merging framework. It is unclear how generation is performed under this setup and whether the method supports true autoregressive decoding at inference time.
2. The evaluation presents perplexity scores across models with different tokenization schemes. Perplexity is not directly comparable across models with different vocabularies (e.g., character-level vs. subword-level). This weakens the strength of the empirical results.
3. The theoretical insights in Section 3.6 are quite minimal and could be significantly expanded to provide more intuition or justification for the method.

## Suggestions:

1. **Related Work:**
The paper does not engage with recent literature on dynamic tokenization, which is directly relevant to the proposed method. In addition to ByT5 and Charformer, the authors should include recent work such as: [Nawrot et al. (2023)](https://aclanthology.org/2022.findings-naacl.117/), [Kallini et al. (2024)](https://openreview.net/forum?id=VYWBMq1L7H), and [Pagnoni et al. (2024)](https://arxiv.org/abs/2412.09871). These works introduce dynamic merging and compression strategies and would help better situate QuickMerge++ in the context of modern alternatives. Section 2.2 is a natural place to incorporate this discussion.
2. **Terminology and Consistency:**
The paper uses both "QuickMerge" and "QuickMerge++" inconsistently. The title and introduction use "QuickMerge++", the abstract uses "QuickMerge", and Section 3 refers to "the original QuickMerge" as though it is a previous paper. Please clarify whether "QuickMerge" is prior work or a baseline ablation in this paper.
3. **Notation Clarity:**
Certain terms are not well defined. What do "X^src" and "X^dst" refer to in Equation (7)? What are "src" and "dst" in this context? Also, the variable "m_i" in Equation (7) is not defined.
All symbols should be defined when introduced to improve readability.
4. **Evaluation Metrics:**
Do not compare perplexities across models with different vocabularies. Instead, use a tokenizer-independent metric such as bits-per-byte, which is more appropriate when comparing character-level and subword-level models.

**Overall Recommendation:**
Reject

**Justification:**
The method proposed is promising and appears to improve on existing approaches to sequence compression in generative models. However, the paper currently suffers from clarity issues—particularly in how the generation process works—and makes inappropriate evaluation comparisons. With revisions to improve clarity and evaluation, this work could be a meaningful contribution.

---

### Decision · Program_Chairs · 2025-06-10

Accept